# Increasing liver stiffness is associated with higher incidence of hepatocellular carcinoma in hepatitis C infection and non-alcoholic fatty liver disease–A population-based study

**Perica Davitkov**[1,2☉], **Kyle Hoffman**[2,3,4☉] *, **Yngve Falck-Ytter**[1,2‡], **Brigid Wilson**[1‡], **Gjorgje Stojadinovikj**[2‡], **Donald D. Anthony**[2,3‡], **Stanley Martin Cohen**[2,5‡], **Gregory Cooper**[2,5‡]

**1** Gastroenterology and Hepatology Section, VA Northeast Ohio Healthcare System, Cleveland, Ohio, United States of America, **2** Case Western Reserve University School of Medicine, Cleveland, Ohio, United States of America, **3** Section of Internal Medicine, VA Northeast Ohio Healthcare System, Cleveland, Ohio, United States of America, **4** Department of Medicine, University Hospitals Cleveland Medical Center / Seidman Cancer Center, Cleveland, Ohio, United States of America, **5** Digestive Health Institute, University Hospitals Cleveland Medical Center / Seidman Cancer Center, Cleveland, Ohio, United States of America

☉ These authors contributed equally to this work.
‡ These authors also contributed equally to this work.
* kyle.hoffman2@uhhospitals.org

## Abstract

### Background & aims

Both non-alcoholic fatty liver disease (NAFLD) and hepatitis C virus (HCV) infection commonly result in hepatic fibrosis and may lead to cirrhosis. This study aims to determine the incidence of HCC in patients with HCV or NAFLD complicated by advanced fibrosis, inferred from measurements of liver stiffness.

### Methods

Using Veterans Affairs (VA) Informatics and Computing Infrastructure (VINCI), we identified a nationwide cohort of patients with an existing diagnosis of HCV or NAFLD with liver transient elastography (TE) testing from 2015 to 2019. HCC cases, along with a random sample of non-HCC patients, were identified and validated, leading to calculation of incidence rates for HCC after adjustment for confounders.

### Results

26,161 patients carried a diagnosis of HCV and 13,629 were diagnosed with NAFLD at the time of testing. In those with HCV, rates of HCC increased with liver stiffness with incidences of 0.28 (95% CI 0.24, 0.34), 0.93 (95% CI 0.72, 1.17), 1.28 (95% CI 0.89, 1.79), and 2.79 (95% CI 2.47, 3.14)/100,000 person years for TE score ranges <9.5 kPa, 9.5–12.5 kPa, 12.5–14.5 kPa and >14.5 kPa, respectively, after a median follow-up of 2.3 years. HCC incidence also increased with higher TE liver stiffness measures in NAFLD after a median follow-up of 1.1 years.

**Data Availability Statement:** All relevant data are within the paper and its Supporting Information files.

**Funding:** UH Cleveland Medical Center, Seidman Cancer Center- small internal award for large database studies. Oracle Grant PTAEO: 17572.01. P0534.xxxxx.49275.

## Conclusion

In this retrospective cohort, the incidence of HCC in HCV and NAFLD increases with higher TE liver stiffness measures, confirming that advanced fibrosis portends risk in viral and non-viral fibrotic liver diseases. Additional comparative studies are needed to determine the optimal cut point of TE liver stiffness to inform HCC screening guidelines and approaches.

## Introduction

In 2018, liver cancer, of which hepatocellular carcinoma (HCC) accounts for more than 75% of cases, was estimated to lead to nearly 800 thousand deaths, making it the fourth most common cause of cancer-related mortality worldwide [1,2]. Globally, the burden of this disease is up- trending with a 75% increase in the incidence of liver cancer from 1990 to 2015 [3]. Concurrently, the overall cancer incidence continues to grow with an expectation of 27 million new cancer cases per year by 2040 [3].

Both NAFLD and HCV commonly result in hepatic fibrosis and, ultimately, can lead to cirrhosis. Largely, this occurs in the context of hepatic inflammation. In that setting, fibrogenesis, or deposition of extracellular matrix materials like collagen, is triggered via cytokines and other mediators, including mechanical stress [4]. Pathology can be visualized directly via liver biopsy, but recently, transient elastography (TE), a non-invasive technology that can measure liver stiffness, has emerged as a tool to examine fibrosis in the liver. Using both ultrasound and elastic waves, a shear elasticity probe is able to quantify a reproducible, operator-independent TE liver stiffness measures [5].

It is known that degree of fibrosis correlates with risk of progression to HCC [6]. As a result, TE has been found to be a useful tool in risk-stratification. For example, the American Gastro-enterology Association (AGA) has suggested that based on diagnostic accuracy studies, patients with chronic HCV and a TE score of <9.5 kilopascals (kPa) who have achieved sustained virologic response after 12 weeks of treatment (SVR12) could be discharged from dedicated liver clinics, as they are considered low risk for development of HCC. However, the magnitude of actual risk that advanced fibrosis without cirrhosis confers is incompletely understood.

Surveillance programs for HCC in high-risk individuals have been proposed and are thought to both add survival benefit and be cost-effective. The European guidelines incorporate early screening for HCC in patients with stage 3 fibrosis (F3), as identified through liver biopsy or TE [7]. Some hepatologists in the USA have adapted the European guidelines and use TE scores above 9.5 kPa, indicative of stage 3 or 4 fibrosis, as an independent indication for screening. In contrast, the AASLD practice guidelines in the United States suggest HCC screening only for patients with cirrhosis, diagnosed either by imaging or clinical parameters suggestive of cirrhosis and its complications, such as ascites, esophageal varices, hepatic encephalopathy, coagulopathy (as evidenced by international normalized ratio (INR) greater than or equal to 1.2 or platelet counts less than 150,000 on two occasions more than one month apart) [8]. This contrast has major implications, as some studies identify early detection of HCC as critical for successful treatment [7,9].

This discrepancy between two of the most influential liver society practice guidelines presents a clinical question that more data may further inform, as currently the degree of risk that advanced fibrosis without cirrhosis portends for HCC is understudied, as is the risk in patients without evidence of fibrosis. The degree of risk in these patients can help determine screening

protocols in this population. This study aims to determine the incidence of HCC in patients with HCV or NAFLD complicated by advanced fibrosis, inferred from measurements of liver stiffness, and, additionally, to identify other clinical predictors of HCC, such as alpha-fetoprotein (AFP), that may aid the decision of whether or not to screen each patient.

## Methods

### Data sources

Veterans Affairs Informatics and Computing Infrastructure (VINCI), a secure workspace facilitating data access and providing analysis tools, was used for the interrogation of national VA data. We obtained IRB approval through the VA Northeast Ohio System Research Office and the VA Innovation and Research Review System. Through VINCI, we accessed Corporate Data Warehouse (CDW) databases to extract patient data. CDW includes data related to approximately 9 million veterans across all 50 states. Available data includes, but is not limited to, International Classification of Disease-Clinical Modification (ICD-CM) codes (9th and 10th revision), International Classification of Disease-Procedural Classification System (ICD-PCS) codes (9th and 10th revision), as well as demographic data and data from the inpatient/outpatient EMRs and inpatient/outpatient pharmacies. Given that data is de-identified and fully anonymized, the IRB committee waived the requirement for informed consent.

### Patient identification

We identified a nationwide VA cohort of all patients with an existing diagnosis of HCV or NAFLD who underwent liver elastography testing between January 1, 2015 and December 31st, 2019. HCV and NAFLD were identified using ICD codes and liver elastography testing was identified based on CPT codes (S1 Table). In both the group with HCV and the group with NAFLD, patients with cholangiocarcinoma and/or cancer with metastases to the liver were excluded from this study. In the NAFLD population, patients were excluded if they also had diagnoses of hepatitis B or HCV. In the HCV population, those who had hepatitis B coinfection were also excluded.

### TE results

TE results were extracted from the liver elastography procedure note or clinical notes using pattern matching and regular expression methods validated on 2,450 subsets, sampled to capture all VA facilities. Not all subjects having liver elastography resulted in a valid TE score. Subjects without numeric score or with a score that was below or above the device readings range 2 to 75 kPa were excluded. However, when compared, there was no difference between the subjects with valid TE score and those without numeric score. For patients with more than one TE score, the follow up period began after the first TE procedure but we included the highest score among all procedures. Specific cutoffs for liver stiffness corresponding to cirrhosis and advanced fibrosis were explored in the latest AGA technical review and guideline on the role of TE in chronic liver diseases. We used these cutoffs to define our population [10]. A TE score of <9.5 kPa was felt to represent a low likelihood of advanced fibrosis while values between 9.5 and 12.5 kPa were felt to represent increased likelihood of advanced fibrosis with low likelihood of cirrhosis. Values between 12.5 and 14.5 kPa represented a high likelihood of advanced fibrosis with some overlap of cirrhosis and, finally, values above 14.5 kPa were felt to represent a high likelihood of cirrhosis.

### HCC case ascertainment

Index HCC cases were identified using inpatient and outpatient ICD codes, as well as a previously described approach using the VA Oncology Domain which contains data abstracted by local cancer registrars using Oncotrax clinical software [11]. Patients with multiple outpatient HCC diagnoses, an inpatient HCC diagnosis, the addition of HCC to a clinical problem list, or an Oncotrax entry were considered cases. In order to exclude potential false positive or "rule out" diagnoses, those with a single instance of an outpatient HCC were excluded. All HCC cases identified in this manner and a random sample of non-HCC patients were validated via detailed chart review.

### Study design

This is a retrospective cohort study using population-based data. For all patients who had liver elastography testing, patients were followed for the development of HCC through 1/8/2020. Data with respect to age, gender, and race, as well as HCV and hepatitis B status, presence or absence of NAFLD, tobacco use, alcohol use, hypertension, dyslipidemia, and diabetes mellitus, was collected at the time of testing. In addition, laboratory data from the time of, or prior to, the liver elastography testing, such as platelets, bilirubin, INR, and AFP, was obtained.

   The two cohorts of patients that had either a diagnosis of NAFLD or HCV prior to elastography testing were followed for the development of HCC. In the group with NAFLD, there was a median follow-up of 1.1 years. In the cohort with HCV, there was a median follow-up of 2.3 years. For each cohort, we calculated patient years at risk at each of the 4 TE score ranges. Next, we calculated the HCC incidence rates for each of the 4 TE score ranges. In addition, among those who developed HCC, the SVR12 status, as well as the temporal association to the diagnosis of HCC, was determined.

### Statistical analysis

Among the patients who underwent liver elastography testing, we summarized our HCV and NAFLD patients undergoing analysis with regards to demographics. For each cohort (NAFLD and HCV), incidence rates of HCC were calculated with exact confidence intervals based on the Poisson distribution. Next, incidence rate ratios with Wald 95% confidence intervals were estimated using Poisson regression. Additional Poisson regression models included age, BMI, diabetes, smoking status, and alcohol abuse diagnosis to estimate incidence rate ratios of TE score intervals after adjusting for these possible confounders. Within these cohorts, the confidence intervals were used to compare incidences for each grouping, organized by TE score. All analysis was performed in R 4.0.2.

## Results

### HCC in all undergoing liver elastography

Liver elastography testing was performed 79,267 times on 65,547 unique patients. Of these, 66,549 tests on 55,365 unique patients had a result identified using pattern matching and regular expressions that were considered in the valid range. We excluded patients with a diagnosis of Hepatitis B at the time of testing and results that indicated a score of <2 and >75 kPa. Among the remaining patients with tests, 26,161 patients had a diagnosis of HCV and 13,629 carried a diagnosis of NAFLD at the time of testing and comprised our study cohort. Within this population, a chart-confirmed diagnosis of HCC was made in 538 patients. Roughly 10% of HCC cases that met database criteria were excluded based on chart review. The majority of these exclusions were deemed to be suspected cholangiocarcinoma upon chart review, despite

**Table 1. Baseline characteristics of the patient populations in HCV and NAFLD analysis populations, categorized by first resolved TE result.**

| | HCV analysis cohort | | NAFLD analysis cohort | |
|---|---|---|---|---|
| | First resolved TE result | | First resolved TE result | |
| | <12.5 kPa | > = 12.5 kPa | <12.5 kPa | > = 12.5 kPa |
| Number of patients | 20647 | 5514 | 10970 | 2659 |
| Age at TE (Years) | 62+/-8.6 | 63.1+/-6.3 | 56.2+/-13.6 | 61.9+/-11.1 |
| Year of TE, median (IQR) | 2017 (2016, 2018) | 2017 (2016, 2018) | 2018 (2018, 2019) | 2018 (2018, 2019) |
| Sex: Male | 19901 (96%) | 5349 (97%) | 9776 (89%) | 2475 (93%) |
| Race: White | 9400 (46%) | 2915 (53%) | 7609 (69%) | 2105 (79%) |
| Race: Black | 10080 (49%) | 2205 (40%) | 2212 (20%) | 317 (12%) |
| Race: Other | 238 (1%) | 77 (1%) | 403 (4%) | 67 (3%) |
| Race: Unknown/missing/declined | 929 (4%) | 317 (6%) | 746 (7%) | 170 (6%) |
| Ethnicity: Hispanic | 903 (4%) | 332 (6%) | 1143 (10%) | 266 (10%) |
| Ethnicity: Non-Hispanic | 19142 (93%) | 4997 (91%) | 9456 (86%) | 2316 (87%) |
| Ethnicity: Unknown/missing/declined | 602 (3%) | 185 (3%) | 371 (3%) | 77 (3%) |
| BMI at TE (kg/m$^2$) | 27.3+/-5.1 | 28.5+/-5.8 | 32.6+/-5.4 | 34.8+/-6.6 |
| Alcohol abuse diagnosis at TE | 5368 (26%) | 1523 (28%) | 1191 (11%) | 374 (14%) |
| Tobacco use diagnosis at TE | 5181 (25%) | 1301 (24%) | 622 (6%) | 176 (7%) |
| Diabetes diagnosis at TE | 4861 (24%) | 1851 (34%) | 3886 (35%) | 1708 (64%) |

Abbreviations: HCV = hepatitis C virus, NAFLD = non-alcoholic fatty liver disease, TE = transient elastography, kPa = kilopascals, BMI = body mass index, kg/m$^2$ = kilograms per meters squared.

the presence of an HCC diagnosis code. Demographic data in patients with HCV and NAFLD compared for TE scores <12.5 and > = 12.5 is presented in Table 1. Comparative demographic date between patients with and without HCC is presented in S2 Table.

The laboratory data from the time of liver elastography testing revealed similar values at baseline between the group who did and did not develop HCC (S3 Table).

## HCC in the HCV subgroup

Among the 26,161 patients with HCV who underwent liver elastography testing, there were 496 patients with chart-confirmed HCC. As demonstrated in Fig 1 and Table 2, in this population, the incidence of HCC increased with TE score with incidence rates of 0.28 (95% CI 0.24 to 0.34), 0.93 (95% CI 0.72 to 1.17), 1.28 (95% CI 0.89 to 1.79), and 2.79 (95% CI 2.47 to 3.14)/ 100,000 person years for TE score range <9.5 kPa, 9.5–12.5 kPa, 12.5–14.5 kPa and >14.5 kPa, respectively. The largest relative increase in risk between adjacent cohorts occurs between the group with TE scores <9.5 and the group with scores between 9.5 and 12.5, with an incidence rate ratio (IRR) of 3.25 (95% CI = (2.42, 4.34)). However, with each increment in terms of TE score, the risk of HCC increased. The IRR for 12.5–14.4 kPa vs. 9.5–12.4 kPa was 1.38 (0.91, 2.06), and the IRR for ≥14.5 kPa vs. 12.5–14.4 kPa was 2.18 (1.55, 3.17). Similar IRRs were observed between TE levels after adjusting for age, smoking status, alcohol abuse, BMI, and diabetes.

Of the 496 patients with HCV who developed HCC, 172 (34.7%) had achieved SVR12 at least 1 year before developing HCC, 184 (37.1%) had SVR12 but developed HCC within one year, and 140 never achieved SVR (28.2%) during the study period. Among 17,299 patients without advanced fibrosis (TE score <9.5), there were a total of 119 HCC cases, of which 47 had achieved SVR12 at least 1 year before developing HCC and 69 either had SVR12 but developed HCC within a year or never achieved SVR12. Incidence of HCC for the patients who

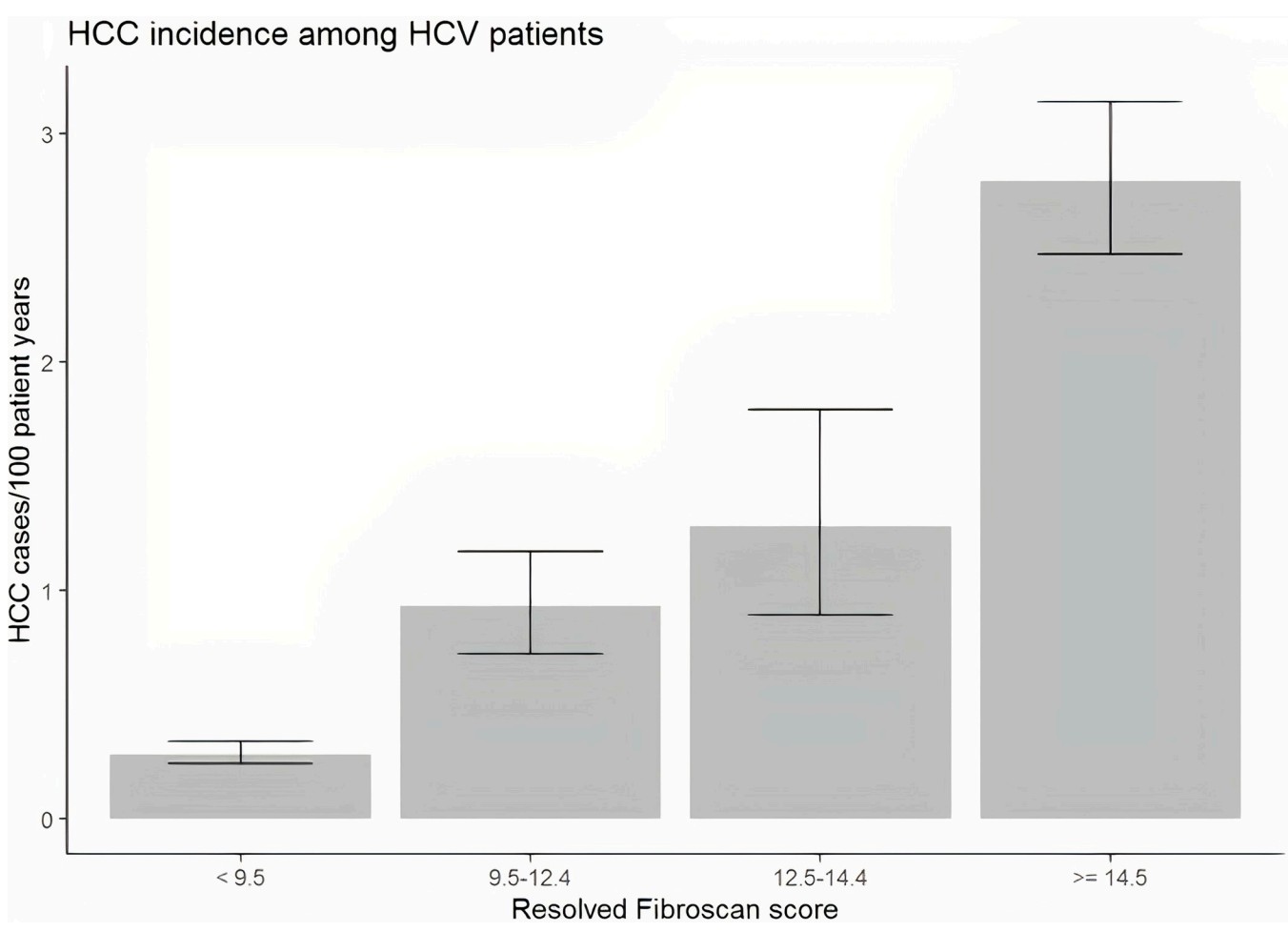

**Fig 1. Incidence of HCC in veteran population with HCV stratified by TE score.**

achieved SVR 12 was 0.117/100 patient years (95% CI: 0.086, 0.156). Those with no SVR 12 had a higher incidence of HCC of 0.171 (0.13, 0.21)/100 patient years.

## HCC in the NAFLD subgroup

Among those tested with liver elastography meeting our inclusion criteria, 13,629 had a diagnosis of NAFLD. Within that population, 42 had chart-review confirmed HCC. Fig 2 displays the incidence of HCC for each cohort of patients based on TE score. For TE scores < 9.5 kPa,

**Table 2. Incidence of HCC in veteran population with HCV based on TE score.**

| Transient elastography (kPa) | Patients with HCC | Patient years at risk | Incidence of HCC per 100 person-years (95% confidence interval) | IRR (95% confidence interval), vs. <9.5 | Adjusted (Age, smoking, ETOH, BMI, DM) IRR |
|---|---|---|---|---|---|
| <9.5 | 119 | 41776 | 0.28 (0.24, 0.34) | Ref. | Ref. |
| 9.5–12.4 | 72 | 7775 | 0.93 (0.72, 1.17) | 3.25 (2.42, 4.34) | 3.25 (2.41, 4.34) |
| 12.5–14.4 | 34 | 2656 | 1.28 (0.89, 1.79) | 4.49 (3.02, 6.50) | 4.35 (2.91, 6.32) |
| ≥14.5 | 271 | 9712 | 2.79 (2.47, 3.14) | 9.80 (7.92, 12.19) | 9.83 (7.93, 12.26) |

Abbreviations: kPa = kilopascals, HCC = hepatocellular carcinoma, IRR = incidence rate ratio, ETOH = alcohol use, BMI = body mass index, DM = diabetes mellitus.

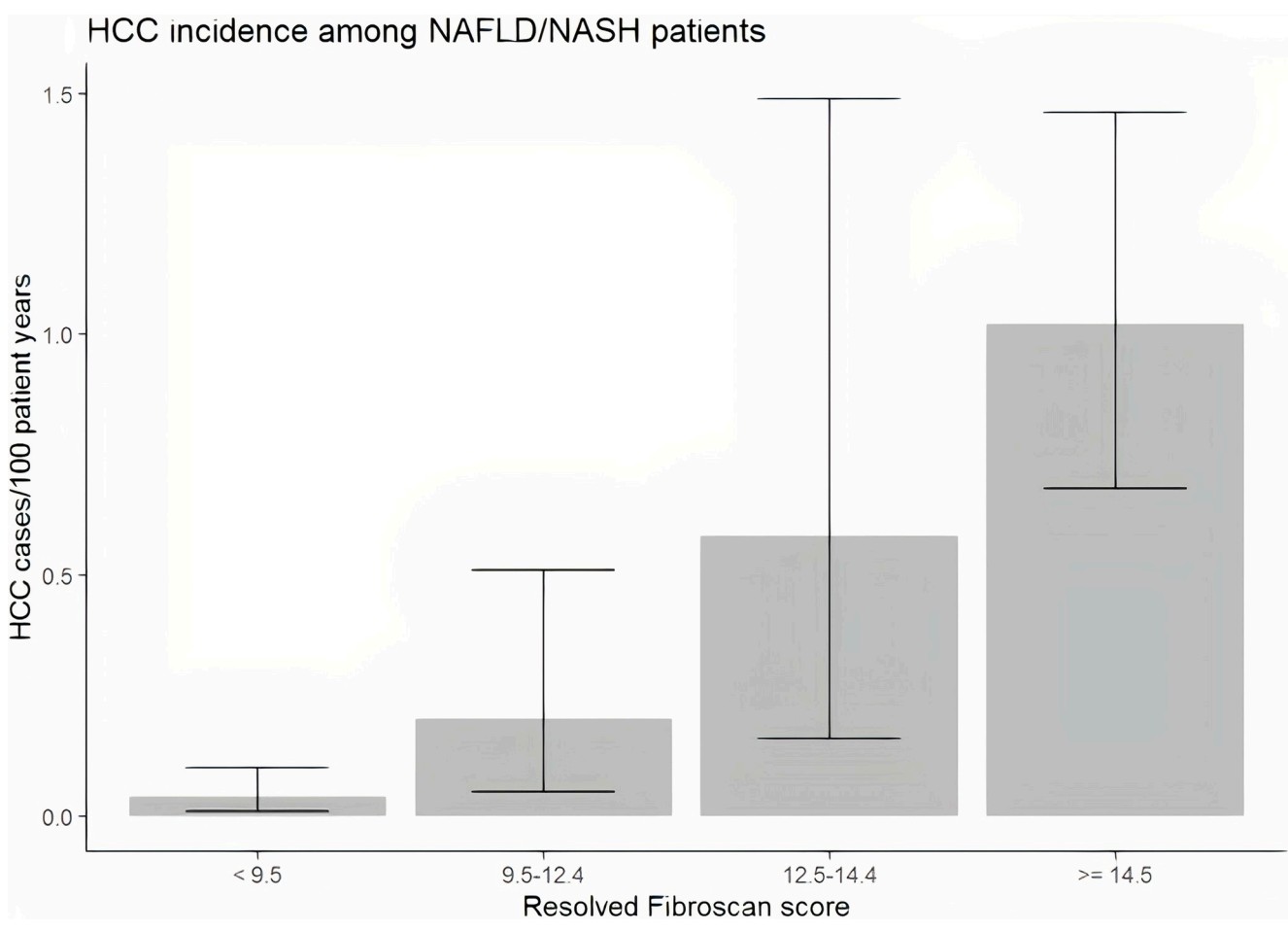

**Fig 2. Incidence of HCC in veteran population with NAFLD stratified by TE score.**

the incidence was 0.04/100 patient years (95% CI 0.01 to 0.1). Relative to this group, the incidence was significantly higher for those with scores between 9.5 kPa and 12.4 kPa (incidence = 0.2, IRR (95% CI) = 4.81 (1.19, 8.20)). For those with TE scores between 12.5 kPa and 14.4 kPa, an incidence of 0.58/100 patient years (95% CI 0.16 to 1.49) was noted (Table 3), yielding an IRR of 2.94 (0.69, 12.42) relative to those with scores between 9.5 kPa and 12.4 kPa. Finally, for a TE score greater than or equal to 14.5, an incidence of 1.02/100 patient years (95% CI 0.68 to 1.46) was found, with an IRR of 1.74 (0.69, 5.88) relative to those with scores

**Table 3. Incidence of HCC in veteran population with NAFLD based on TE score.**

| Transient elastography (kPa) | Patients with HCC | Patient years at risk | Incidence of HCC per 100 person-years (95% confidence interval) | IRR (95% confidence interval), vs. <9.5 | Adjusted (Age, smoking, ETOH, BMI, DM) IRR |
|---|---|---|---|---|---|
| <9.5 | 5 | 12130 | 0.04 (0.01, 0.1) | Ref. | Ref. |
| 9.5–12.4 | 4 | 2015 | 0.2 (0.05, 0.51) | 4.81 (1.19, 18.20) | 3.87 (0.95, 14.68) |
| 12.5–14.4 | 4 | 686 | 0.58 (0.16, 1.49) | 14.14 (3.50, 53.46) | 9.99 (2.45, 38.15) |
| ≥14.5 | 29 | 2852 | 1.02 (0.68, 1.46) | 24.67 (10.42, 72.51) | 15.74 (6.45, 47.25) |

Abbreviations: kPa = kilopascals, HCC = hepatocellular carcinoma, IRR = incidence rate ratio, ETOH = alcohol use, BMI = body mass index, DM = diabetes mellitus.

between 12.5 kPa and 14.4 kPa. Reduced but consistent IRRs were observed by increasing TE levels after adjusting for age, smoking status, alcohol abuse, BMI, and diabetes.

## Discussion

### Predicting HCC development

Accurate prediction of HCC risk is vitally important to the development of successful and cost-effective screening practices. In this study of a large veteran population with chronic liver disease who have undergone TE testing, we found that patients with advanced fibrosis and HCV or NAFLD were at increased risk for HCC. The findings suggest that, in contrast to AASLD guidelines, screening this subgroup of patients may be prudent.

HCV infection was a significant risk factor for the development of HCC and had higher incidence rates when compared to NAFLD cohort. Previous studies have also suggested that those with NAFLD who develop HCC are more likely to present at an older age and our cohort may not have included a sufficient number of elderly patients and did not have an adequate follow up time in the NAFLD cohort [12]. In addition, there are differences in natural progression, with HCV leading to cirrhosis in up to 20% of patients [13]. In contrast, the rate of cirrhosis reported in NAFLD (ranging in severity from type 1 to type 4) was approximately 15% over more than 8 years of follow-up [14]. However, it may also suggest that there may be a lower incidence of advanced fibrosis or cirrhosis in the NAFLD population, as compared to HCV, as well as the potential that HCV has direct oncogenic properties independent of the development of fibrosis or cirrhosis.

We further explored other risk factors, such as age, gender, race ethnicity, alcohol, tobacco, diabetes mellitus, hypertension, and BMI. In comparison between low and high TE score for the HCV and NAFLD cohorts, there was no substantial difference in age, gender race and ethnicity. Although the study was not designed to detect difference in developing HCC between different gender, race, and ethnicity, our results of higher HCC rates in males and Black race are consistent with the previous studies describing nearly 2.5 times the amount of incident cases of HCC in men when compared to women and incidence rate of 4.2 per 100,000 persons in Blacks as compared to 2.6 in whites [15,16]. Again, this is a different population than that studied here, with our population representing those undergoing TE testing and the referenced study representing the population at large. However, similar trends are seen.

Laboratory data revealed similar values at baseline between the groups who did and did not develop HCC. No significant differences were seen, though there may have been a trend towards lower platelets and higher AFP. The degree of similarity suggests that laboratory data may not be particularly useful in determining whom to screen unless it suggests cirrhosis, which is a known risk factor for HCC development.

### HCC in the HCV subgroup

Though rates of new HCV-related chronic liver disease declined from the late 1980's to the early 2010's, recent studies have suggested a steady increase in incidence of HCV infection since 2010 with an estimated 50,300 acute infections in 2018 [17,18]. This increase has largely been attributed to resurgence in intravenous drug use in the setting of the ongoing opioid epidemic [18].

With a median follow-up of 2.3 years, the incidence for HCC in those with HCV was 0.82% per year. Fig 1 shows an increasing incidence of HCC with increasing TE. Previously, it was known that cirrhosis and advanced fibrosis were risk factors for HCC, though the degree of risk that advanced fibrosis led to was not fully understood [6]. This study was able to quantify how increasing TE scores directly correlate with increased risk of HCC and, thus, may be used

to help determine who to screen for HCC. The incidence of 2.79 per 100 person-years in the > 14.5 kPa group in our study is consistent with published risk (3.7% per 100 person years in those with HCV-related cirrhosis in Europe and the United States) [19]. However, we are also presenting a substantial risk for developing HCC with incidences of 0.93 and 1.28 per 100 person-years in the advanced fibrosis groups.

Furthermore, those who developed HCC in the context of HCV were at varying stages of treatment and virologic response. In those with cirrhosis, antiviral therapy has been shown to reduce the risk of the development of HCC [20]. Interestingly, in our population, a majority of patients had already achieved SVR at the time of diagnosis of HCC. Previous studies have shown that many patients with HCV continue to have elevated mean fibrosis scores [21]. This suggests that any potential screening intervention should continue independent of SVR status in those with increased liver stiffness. However, serial measurements of TE scores may be of utility for quantification of further risk after SVR, as there is evidence that some regression of fibrosis may occur after SVR and, thus, lower the risk of HCC [22]. Inaddition, we found that the incidence of HCC in the individuals who clear the HCV virus and are without advanced fibrosis (TE <9.5) is very low 1 in 1000 person-years. Thus, it is unlikely that HCC surveillance in this risk group will be cost effective. However, further studies should validate these findings and perform detailed cost-effectiveness analysis.

Lastly, patients with high TE scores in the HCV cohort were more likely to have DM when compared to the patients with low TE scores, but all the other demographic variables were similar. However, we did adjust our analysis for age, smoking, alcohol BMI and DM.

## HCC in the NAFLD subgroup

Since 1988, the rates of NAFLD have been steadily increasing as the US population continues to age and rates of comorbidities, such as diabetes, hypertension, and obesity, rise [17]. Between 2013 and 2016, the prevalence of NAFLD was estimated at 31.9%, as compared to a prevalence of 20% between 1988 and 1994 [17].

With a median follow-up of 1.1 years, the incidence of HCC was 0.23% per year in those with NAFLD. When compared with the cohorts of HCV patients with equivalent TE scores, the incidence rates were significantly lower for all groups of TE scores except those at risk of having established cirrhosis (between 12.5 and 14.5 kPa). However, the significantly lower incidence rates across the advanced fibrosis groups when compared to the HCV cohort need to be further validated.

However, similar to HCV, with each increase in the stratification of TE score, there was a trend towards higher risk of HCC with all such IRR exceeding 1.7. TE scores equal to or above 14.5 kPa were associated with a significantly higher risk than the group with scores between 9.5 and 12.4 or less than 9.5 kPa. Similarly, the group with TE scores between 12.5 and 14.4 were associated with a higher incidence of HCC, as compared to the group with TE scores less than 9.5. In NAFLD cohort, there was a trend to a higher BMI and more diabetes in patients in the higher TE score group when compared to the group with lower TE scores. Again, the adjusted analysis suggests that both advanced fibrosis and cirrhosis are risk factors for HCC in NAFLD.

In the future, a detailed cost-effectiveness analysis may show that TE is not as useful in NAFLD and/or it may not be warranted to screen for HCC in NAFLD patients with advanced fibrosis.

## Strengths

This population-based study enrolled a large number of patients to accurately inform the risk of HCC development in HCV and NAFLD at various levels of fibrosis as measured non-

invasively with transient elastography. Confounding risk factors were carefully identified, and manual chart review was carried out to validate data extraction. Also, an adjusted analysis for confounding factors was performed.

## Limitations

This study was conducted in the predominantly male veteran population and excluded those with conditions such as HBV superinfection, and those with simultaneous HCV and NAFLD These factors may limit the generalizability of the study. Furthermore, this study relied on accurate coding for diagnoses and testing as well as accuracy of reported liver stiffness values. Also, some patients may have sought care at facilities outside of the Veterans Affairs network and, thus, be lost to follow-up in this study. Finally, short follow-up period which limits the number of HCC's that developed.

## Future directions

Further elucidation and validation of the relationship between TE scores and the development of HCC could be used to develop screening protocols. In addition, quantification of the risk, based on TE scores and other factors, such as demographics, could make any potential screening intervention a cost-effective intervention. However, any screening intervention should be studied directly for its effects on HCC prevention and mortality reduction, as well as harms, with appropriate methods (i.e. RCTs), including concurrent accurate collection of resource use data.

## Conclusion

In this large retrospective cohort, we found that the incidence of HCC in both HCV and NAFLD escalates with increasing TE score, confirming that advanced fibrosis, not just cirrhosis, portends risk. Further studies validating and further elucidating this relationship between the development of HCC and TE scores would be useful in informing current HCC screening guidelines and in the development of new HCC screening protocols.

## Supporting information

**S1 Table. ICD and CPT codes used for patient identification.**
(DOCX)

**S2 Table. Baseline demographic data in those who did and did not develop HCC.**
(DOCX)

**S3 Table. Baseline laboratory data in those who did and did not develop HCC.**
(DOCX)

## Acknowledgments

The authors would like to acknowledge the support of Case Western Reserve University, University Hospitals and Seidman Cancer Center, and the Louis Stokes VA Medical Center.

## Author Contributions

**Conceptualization:** Perica Davitkov, Yngve Falck-Ytter, Brigid Wilson.

**Data curation:** Perica Davitkov, Brigid Wilson, Gjorgje Stojadinovikj, Donald D. Anthony, Stanley Martin Cohen.

**Formal analysis:** Perica Davitkov, Kyle Hoffman, Yngve Falck-Ytter, Brigid Wilson, Gjorgje Stojadinovikj, Donald D. Anthony, Stanley Martin Cohen, Gregory Cooper.

**Funding acquisition:** Perica Davitkov.

**Investigation:** Perica Davitkov, Kyle Hoffman, Stanley Martin Cohen.

**Methodology:** Perica Davitkov, Gregory Cooper.

**Supervision:** Yngve Falck-Ytter, Stanley Martin Cohen, Gregory Cooper.

**Writing – original draft:** Perica Davitkov, Kyle Hoffman.

**Writing – review & editing:** Perica Davitkov, Kyle Hoffman, Yngve Falck-Ytter, Brigid Wilson, Gjorgje Stojadinovikj, Donald D. Anthony, Stanley Martin Cohen, Gregory Cooper.

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
