## [Decision Letter · Decision Letter 0]

26 Oct 2022

PONE-D-22-13686Increasing liver stiffness is associated with higher incidence of hepatocellular carcinoma – a population-based studyPLOS ONE

Dear Dr. Hoffman,

Thank you for submitting your manuscript to PLOS ONE. After careful consideration, we feel that it has merit but does not fully meet PLOS ONE’s publication criteria as it currently stands. Therefore, we invite you to submit a revised version of the manuscript that addresses the points raised during the review process.

ACADEMIC EDITOR:Your manuscript has been assessed by our reviewers. They have raised a number of points which we believe would improve the manuscript and may allow a revised version to be published in PLOS ONE.

We look forward to receiving your revised manuscript.

Kind regards,

Yuh-Shin Chang, MD, PhD

Academic Editor

PLOS ONE

Journal Requirements:

3. In ethics statement in the manuscript and in the online submission form, please provide additional information about the patient records/samples used in your retrospective study. Specifically, please ensure that you have discussed whether all data/samples were fully anonymized before you accessed them and/or whether the IRB or ethics committee waived the requirement for informed consent. If patients provided informed written consent to have data/samples from their medical records used in research, please include this information.

None of the authors reports conflict of interest 

Reviewers' comments:

Reviewer's Responses to Questions

**Comments to the Author**

1. Is the manuscript technically sound, and do the data support the conclusions?

Reviewer #1: Yes

Reviewer #2: Yes

Reviewer #3: Yes

2. Has the statistical analysis been performed appropriately and rigorously? 

Reviewer #1: Yes

Reviewer #2: Yes

Reviewer #3: Yes

3. Have the authors made all data underlying the findings in their manuscript fully available?

Reviewer #1: Yes

Reviewer #2: Yes

Reviewer #3: Yes

4. Is the manuscript presented in an intelligible fashion and written in standard English?

Reviewer #1: Yes

Reviewer #2: Yes

Reviewer #3: No

5. Review Comments to the Author

Reviewer #1: This is a retrospective study assessing TE stiffness relation with HCC in patients with HCV or NASH. The following suggestions should be considered:

1. In the abstract it should be stated the duration of the mean followed up for patients.

2. To include p-value for characteristics difference in table 1.

3. To include a parapragh on the effect of HCV treatment on the population as part of the discussion.

Reviewer #2: The authors analyzed the rates of HCC in both HCV and NAFLD patients. They concluded that HCC incidence increased with higher TE liver stiffness measures in both HCV and NAFLD. The experiments were conducted properly. The conclusions was solidly based on the data analyzed.

Reviewer #3: In this manuscript, Davitkov et al. present the findings of increased incidence of hepatocellular carcinoma in non-alcoholic fatty liver disease (NAFLD) and hepatitis C virus (HCV) infected patients. Overall, the paper is well written and grammatically correct. However, the clear rationale for conducting the study and the novelty is lacking. There are concerns about the results which require addressing before the manuscript can be considered for publication.

Results:

1. In Table 1, there are too many gridlines in the table. For example, you have divided the heading “Race” into 4 rows including white, black, other, and unknown. You need to hide the gridlines between each group, indent, and merge the 4 subheadings into one row. This will make the table easier for readers to read. Moreover, the abbreviations in the table should be defined in the footnotes.

2. If I am not mistaken, your study population is divided into 2 study groups, HCV and NAFLD. Increased transient elastography was shown, indicating an increased incidence of HCC.

Do you have data which shows that patients with both HCV and NAFLD have a higher incidence rate ratio for developing HCC?

3. Considering your study results, I recommend changing your research title by adding both HCV and NAFLD to your article title. This will enable readers to easily understand why liver stiffness can result in HCC.

6. PLOS authors have the option to publish the peer review history of their article (what does this mean?). If published, this will include your full peer review and any attached files.

Reviewer #1: No

Reviewer #2: No

Reviewer #3: No

---

## [Author Response · Author response to Decision Letter 0]

29 Nov 2022

Updated style and file naming to meet PLOS one requirements.

Changed within the body of the manuscript to reflect approval through the VA Northeast Ohio System Research Office and the VA Innovation and Research Review System.

3. In ethics statement in the manuscript and in the online submission form, please provide additional information about the patient records/samples used in your retrospective study. Specifically, please ensure that you have discussed whether all data/samples were fully anonymized before you accessed them and/or whether the IRB or ethics committee waived the requirement for informed consent. If patients provided informed written consent to have data/samples from their medical records used in research, please include this information.

Data was collected and stored with full Protected Health Information (PHI) on VA servers behind a VA fire wall. For data analysis, identifiers were removed and only coded data was analyzed. Given that data is de-identified and fully anonymized, the IRB committee waived the requirement for informed consent and HIPPA Authorization. The manuscript was updated to reflect this. 

Grant info: UH Cleveland Medical Center, Seidman Cancer Center- small internal award for large database studies. Oracle Grant PTAEO: 17572.01.P0534.xxxxx.49275. When we resubmit, the funding info/financial disclosures will all reflect this.

None of the authors reports conflict of interest 

This was added to the cover letter in a competing interests section.

Added.

Reviewers' comments:

Reviewer's Responses to Questions

Comments to the Author

1. Is the manuscript technically sound, and do the data support the conclusions?

Reviewer #1: Yes

Reviewer #2: Yes

Reviewer #3: Yes

2. Has the statistical analysis been performed appropriately and rigorously? 

Reviewer #1: Yes

Reviewer #2: Yes

Reviewer #3: Yes

3. Have the authors made all data underlying the findings in their manuscript fully available?

Reviewer #1: Yes

Reviewer #2: Yes

Reviewer #3: Yes

4. Is the manuscript presented in an intelligible fashion and written in standard English?

Reviewer #1: Yes

Reviewer #2: Yes

Reviewer #3: No

Not sure of the specific concerns from reviewer #3 but any further concerns addressed below.

5. Review Comments to the Author

Reviewer #1: This is a retrospective study assessing TE stiffness relation with HCC in patients with HCV or NASH. The following suggestions should be considered:

1. In the abstract it should be stated the duration of the mean followed up for patients.

Added to abstract. 

2. To include p-value for characteristics difference in table 1.

P values added to table 1. 

3. To include a paragraph on the effect of HCV treatment on the population as part of the discussion.

In the third paragraph of the discussion section “HCC in the HCV subgroup”, we discuss the potential effects of HCV treatment on fibrosis. We also added a line about how treatment of HCV has been previously shown to reduce the risk of HCC and corelate with our results. 

Reviewer #2: The authors analyzed the rates of HCC in both HCV and NAFLD patients. They concluded that HCC incidence increased with higher TE liver stiffness measures in both HCV and NAFLD. The experiments were conducted properly. The conclusions was solidly based on the data analyzed.

Reviewer #3: In this manuscript, Davitkov et al. present the findings of increased incidence of hepatocellular carcinoma in non-alcoholic fatty liver disease (NAFLD) and hepatitis C virus (HCV) infected patients. Overall, the paper is well written and grammatically correct. However, the clear rationale for conducting the study and the novelty is lacking. There are concerns about the results which require addressing before the manuscript can be considered for publication.

The rationale for the study is, to date, there is not data on the risk for HCC (and thus whether surveillance is needed) in non-cirrhotic patients with advanced fibrosis. This was made more clear in the revised manuscript.

Results:

1. In Table 1, there are too many gridlines in the table. For example, you have divided the heading “Race” into 4 rows including white, black, other, and unknown. You need to hide the gridlines between each group, indent, and merge the 4 subheadings into one row. This will make the table easier for readers to read. Moreover, the abbreviations in the table should be defined in the footnotes.

Fixed gridlines to make table appear cleaner. Abbreviations now in legend. 

2. If I am not mistaken, your study population is divided into 2 study groups, HCV and NAFLD. Increased transient elastography was shown, indicating an increased incidence of HCC.

Do you have data which shows that patients with both HCV and NAFLD have a higher incidence rate ratio for developing HCC?

This data was not collected – in the NAFLD population, those with HCV were excluded. Though no specific exclusion was made for those with hepatitis C who may have had concurrent NAFLD, no specific data on patients with both pathologies was collected. 

3. Considering your study results, I recommend changing your research title by adding both HCV and NAFLD to your article title. This will enable readers to easily understand why liver stiffness can result in HCC.

Added.

---

## [Decision Letter · Decision Letter 1]

27 Dec 2022

PONE-D-22-13686R1Increasing liver stiffness is associated with higher incidence of hepatocellular carcinoma in hepatitis C infection and non-alcoholic fatty liver disease – a population-based study

PLOS ONE

Dear Dr. Hoffman,

Thank you for submitting your manuscript to PLOS ONE. After careful consideration, we feel that it has merit but does not fully meet PLOS ONE’s publication criteria as it currently stands. Therefore, we invite you to submit a revised version of the manuscript that addresses the points raised during the review process.

We look forward to receiving your revised manuscript.

Kind regards,

Yuh-Shin Chang, MD, PhD

Academic Editor

PLOS ONE

Additional Editor Comments (if provided):

Thank you for this expeditious revision. Your manuscript has been assessed by our reviewers. They have raised many points which we believe would improve the manuscript and may allow a revised version to be published.

Reviewers' comments:

Reviewer's Responses to Questions

**Comments to the Author**

1. If the authors have adequately addressed your comments raised in a previous round of review and you feel that this manuscript is now acceptable for publication, you may indicate that here to bypass the “Comments to the Author” section, enter your conflict of interest statement in the “Confidential to Editor” section, and submit your "Accept" recommendation.

Reviewer #1: All comments have been addressed

Reviewer #3: (No Response)

2. Is the manuscript technically sound, and do the data support the conclusions?

Reviewer #1: Yes

Reviewer #3: Partly

3. Has the statistical analysis been performed appropriately and rigorously? 

Reviewer #1: Yes

Reviewer #3: Yes

4. Have the authors made all data underlying the findings in their manuscript fully available?

Reviewer #1: Yes

Reviewer #3: Yes

5. Is the manuscript presented in an intelligible fashion and written in standard English?

Reviewer #1: Yes

Reviewer #3: No

6. Review Comments to the Author

Reviewer #1: Thanks to the authors for following previous suggestions. All my previous comments have been addressed in the current draft.

Reviewer #3: Results:

1. In Table 1, you put the description of abbreviations and statistical methods in the title. Please transfer this part to the footnote following Table 1. Besides, in Table 1, you missed the units for many variables. For example, “Age at TE” should be “Age at TE (years).”

2. In Table 2, please spell out the terms that the abbreviations stand for (ETOH, BMI, DM, etc.) in the footnote.

3. In addition, we would appreciate your responses indicated by “Reply: Thanks for your suggestions or Thanks for ……” Otherwise, we cannot easily recognize where your point-by-point answers begin.

4. In your supplementary Table S1, what does “%” indicate in “ICD9s for HCV: '070.41%…’”? ICD9 or ICD10 codes do not use the % symbol.

5. Anyway, I hope you can improve the editing quality of the table or figure design. I suggest reviewing some previous published articles for examples.

7. PLOS authors have the option to publish the peer review history of their article (what does this mean?). If published, this will include your full peer review and any attached files.

Reviewer #1: No

Reviewer #3: No

---

## [Author Response · Author response to Decision Letter 1]

29 Dec 2022

Reviewer #1: Thanks to the authors for following previous suggestions. All my previous comments have been addressed in the current draft.

Reviewer #3: Results:

1. In Table 1, you put the description of abbreviations and statistical methods in the title. Please transfer this part to the footnote following Table 1. Besides, in Table 1, you missed the units for many variables. For example, “Age at TE” should be “Age at TE (years).”

Reply: Thanks for your suggestion. We have moved the abbreviations to a footnote following table 1. In the footnote, we also added explanations of abbreviations: kPa and kg/m2. In addition, we added units to “Age at TE” and “BMI at TE”. 

2. In Table 2, please spell out the terms that the abbreviations stand for (ETOH, BMI, DM, etc.) in the footnote.

Reply: Thanks for your suggestion. We added a footnote following table 2 with abbreviations. 

3. In addition, we would appreciate your responses indicated by “Reply: Thanks for your suggestions or Thanks for ……” Otherwise, we cannot easily recognize where your point-by-point answers begin.

Reply: Thanks for this note. All replies to this revision request are formatted this way. 

4. In your supplementary Table S1, what does “%” indicate in “ICD9s for HCV: '070.41%…’”? ICD9 or ICD10 codes do not use the % symbol.

Reply: Thanks for your suggestion. We have removed the percent symbols – they were likely a vestige of the way the data was extracted from the database. 

5. Anyway, I hope you can improve the editing quality of the table or figure design. I suggest reviewing some previous published articles for examples

Reply: Thanks for your comment. I reviewed recent articles and changed the formatting/text of all of the tables. As far as the figures, I’m not sure what would increase the editing quality, but happy to adjust anything as needed!

---

## [Decision Letter · Decision Letter 2]

5 Jan 2023

Increasing liver stiffness is associated with higher incidence of hepatocellular carcinoma in hepatitis C infection and non-alcoholic fatty liver disease – a population-based study

PONE-D-22-13686R2

Dear Dr. Hoffman,

We’re pleased to inform you that your manuscript has been judged scientifically suitable for publication and will be formally accepted for publication once it meets all outstanding technical requirements.

Kind regards,

Yuh-Shin Chang, MD, PhD

Academic Editor

PLOS ONE

Additional Editor Comments (optional):

Reviewers' comments:

Reviewer's Responses to Questions

**Comments to the Author**

1. If the authors have adequately addressed your comments raised in a previous round of review and you feel that this manuscript is now acceptable for publication, you may indicate that here to bypass the “Comments to the Author” section, enter your conflict of interest statement in the “Confidential to Editor” section, and submit your "Accept" recommendation.

Reviewer #3: All comments have been addressed

2. Is the manuscript technically sound, and do the data support the conclusions?

Reviewer #3: Partly

3. Has the statistical analysis been performed appropriately and rigorously? 

Reviewer #3: Yes

4. Have the authors made all data underlying the findings in their manuscript fully available?

Reviewer #3: Yes

5. Is the manuscript presented in an intelligible fashion and written in standard English?

Reviewer #3: Yes

6. Review Comments to the Author

Reviewer #3: Most parts of my concerns have been addressed in the current draft. The authors should learn more from experienced researchers or published articles to improve the presentation of data and the quality of tables or figures.

7. PLOS authors have the option to publish the peer review history of their article (what does this mean?). If published, this will include your full peer review and any attached files.

Reviewer #3: No

---

## [Editor Report · Acceptance letter]

10 Jan 2023

PONE-D-22-13686R2 

Increasing liver stiffness is associated with higher incidence of hepatocellular carcinoma in hepatitis C infection and non-alcoholic fatty liver disease – a population-based study 

Dear Dr. Hoffman:

I'm pleased to inform you that your manuscript has been deemed suitable for publication in PLOS ONE. Congratulations! Your manuscript is now with our production department. 

Kind regards, 

on behalf of

Dr. Yuh-Shin Chang 

Academic Editor

PLOS ONE